# Plasma transport into the duskside magnetopause caused by Kelvin-Helmholtz vortices in response to the northward turning of the interplanetary magnetic field observed by THEMIS

Guang Qing Yan[1], George K. Parks[2], Chun Lin Cai[1], Tao. Chen[1], James P. McFadden[2], Yong Ren[1,3]

[1] State Key Laboratory of Space Weather, National Space Science Center, Chinese Academy of Sciences, Beijing, China, 100190

[2] Space Science Laboratory, University of California, Berkeley, California, USA, CA94720

[3] University of Chinese Academy of Sciences, Beijing, China, 100049

**Abstract: A train of likely Kelvin-Helmholtz (K-H) vortices with plasma transport across the magnetopause has been observed by the Time History of Events and Macroscale Interactions during Substorms (THEMIS) at the duskside of magnetopause. This unique event occurs when the interplanetary magnetic field (IMF) abruptly turns northward, which is the immediate change to facilitate the K-H instability. Two THEMIS spacecraft, TH-A and TH-E, separated by 3 Re, periodically encountered the duskside magnetopause and the low-latitude boundary layer (LLBL) with a period of 2 minutes and tailward propagation of 212 km/s. Despite that surface waves could also explain some of the observations, the rotations in the bulk velocity observation, distorted magnetopause with plasma parameter fluctuations and the magnetic field perturbations, as well as high-velocity low-density feature, indicate the possible formation of rolled-up K-H vortices at the duskside of magnetopause. The coexistence of magnetosheath ions with magnetospheric ions and enhanced energy flux of hot electrons is identified in the K-H vortices. These transport regions appear more periodic at the upstream spacecraft and more dispersive at the downstream location, indicating a significant transport can occur and evolve during the tailward propagation of the K-H waves. There is still much work to do to fully understand the Kelvin-Helmholtz mechanism. The observations of the direct response to the northward turning of the IMF, the possible evidence of plasma transport within the vortices, involving both ion and electron fluxes can provide additional clues to the K-H mechanism.**

**Key words**: K-H vortices, northward IMF, plasma transport, LLBL
**1 Introduction**
Kelvin-Helmholtz (K-H) instability can be activated at the interface between different plasma
regimes with different velocities, and the perturbations propagate along the direction of the
velocity shear as a form of surface wave developing into nonlinear vortices. As shown by
Hasegawa (1975), the high density and the magnetic field perpendicular to the velocity shear on
either side of the interface facilitate the unstable condition. The fastest K-H instability occurs
when the wave vector $k$ is parallel/antiparallel to the velocity shear and perpendicular to the
magnetic field (Southwood, 1979; Manuel & Samson, 1993). This condition favors the
low-latitude magnetopause where the velocity shear and the northward magnetospheric magnetic
field are available. The magnetic tension stabilizes the shear layer if the magnetic field and the
velocity shear are aligned, indicating that the radial IMF does not favor the K-H instability.
However, reported observation indicates that K-H waves occur at the high-latitude magnetopause
under the dawnward IMF and continues to exist when the IMF turns radial (Hwang et al., 2012).
On the other hand, under the radial IMF, K-H instability is found in both simulations (Tang et al.,
2013; Adamson et al., 2016) and observations (Farrugia et al., 2014; Grygorov et al., 2016). In
some cases, the K-H instability is thought facilitated by a denser boundary layer formed by the
dayside magnetic reconnections (Grygorov et al., 2016), by the plasma plume (Walsh et al., 2015),
or by the pre-existing denser boundary layer formed by the high-latitude reconnections under the
northward IMF (Hasegawa et al., 2009; Nakamura et al. 2017). Theoretically, both northward and
southward IMF can favor the K-H instability at the low-latitude magnetopause. In fact, almost all
of the previous observations (Chen & Kivelson, 1993; Kivelson & Chen, 1995; Fujimoto et al.,
2003; Hasegawa et al., 2004) and simulations (Chen et al., 1997; Farrugia et al., 2003; Miura,
1995; Hashimoto & Fujimoto, 2005) show that the K-H waves occur preferentially under the
northward IMF, although linear K-H waves are observed under the southward IMF (Mozer et al.,
1994; Kawano et al., 1994). However, under the southward IMF, Cluster has observed nonlinear
K-H waves with irregular and turbulent characteristics (Hwang et al., 2011) and THEMIS has
observed regular K-H vortices with an induced electric field at the edges (Yan et al., 2014). As
reviewed (Johnson et al., 2014; Masson & Nykyri, 2016) recently, observations from many
missions such as Cluster, THEMIS, Wind, Geotail etc., as well as simulations greatly enriched our
understandings of the K-H instability and the vortices. Based on long term observations, a
statistical survey indicates that K-H waves are much more ubiquitous than previously thought
(Kavosi & Raeder, 2015), which implies the importance of the solar wind plasma transport into
the magnetosphere via the K-H vortices.
In addition to magnetic reconnections at low latitude (Dungey, 1961) and high latitude
magnetopause (Song & Russell, 1992),  whose nature is a popular research topic (e.g., Dai, 2009;
Dai et al., 2017; Dai, 2018), the K-H instability is an important way to transport solar wind into
the magnetosphere when reconnections are inactive at the magnetopause. A statistical study of
Double Star observations implies the entry of cold ions into the flank magnetopause caused by the
K-H vortices that is enhanced by solar wind speed (Yan et al., 2005). However, it is noted that the
K-H instability itself cannot lead to plasma transport across the magnetopause (Hasegawa et al.,
2004); therefore, certain secondary processes (e.g., Nakamura et al., 2004; Matsumoto & Hoshino,

2004; Chaston et al., 2007) are necessarily coupled with the K-H instability for plasma transport into the magnetosphere via the LLBL. The reconnection of the twisted magnetic field lines inside the K-H vortex was first found in a simulation (Otto & Fairfield, 2000) and has since been identified in observations (Nykyri et al., 2006; Hasegawa et al., 2009; Li, et al., 2016). The plasma transport into the magnetosphere via such a process in K-H vortices has been quantitatively investigated in a simulation (Nykyri & Otto, 2001). Most recently, energy transport from a K-H wave into a magnetosonic wave was estimated conserving energy in the cross-scale process, and three possible ways were discussed to transfer energy involving shell-like ion distributions, kinetic Alfvén waves, and magnetic reconnection (Moore et al., 2016). Up to now, reports of direct observations of plasma transport in the K-H vortices are only a hand full (e.g., Sckopke et al., 1981; Fujimoto et al., 1998; Hasegawa et al., 2004). Moreover, the microphysical processes for the plasma transport remains unclear, indicating more observations of such a transport process are needed to help us understand the physics. In this work, we present the THEMIS observations of likely K-H vortices activated when the IMF abruptly turns northward. We show a solar wind transport into the magnetosphere occurs and evolves within the vortices.

## 2 Data and Methods

The THEMIS mission (Angelopoulos, 2008) consists of five identical spacecraft originally orbiting the Earth similar to a string of pearls configuration. In August 2009, TH-B and TH-C were pushed to the vicinity of the lunar orbit, while the other three stayed in the near-Earth orbit with an apogee of approximately 13 Re. The instruments onboard include a flux gate magnetometer (FGM) (Auster et al., 2008) to measure the magnetic field and an electrostatic analyzer (ESA) (McFadden et al., 2008) to measure the electron (6 eV-30 keV) and ion (5 eV-25 keV) fluxes. We used the 3-second averaged FGM and ESA data from TH-A and TH-E to perform the particle analysis, and the 1/16 second averaged FGM data to perform the minimum variance analysis (MVA) (Sonnerup & Cahill, 1967; 1968) to determine the local magnetopause coordinates to find the distortions of the magnetopause. The FGM and ESA data from TH-B located in the dawnside downstream solar wind provide the IMF and solar wind conditions with an estimated time lag of 10 minutes from the subsolar magnetopause to TH-B. Both ion and electron energy spectra with a 3-second resolution were used to diagnose the transport of the magnetosheath and magnetospheric ions. During the interval of interest, there are no data in the top energy channels centered at 25.21 keV for the ion spectrum and 31.76 keV for the electron spectrum, which has not influenced our investigations.

## 3 Observations and Discussions

During the interval UT 22:20-22:54 on March 28, 2016, TH-A and TH-E were located near the magnetopause (figure 1), while TH-D was located in the inner magnetosphere, far from the magnetopause. TH-B, near the lunar orbit, was immersed in the solar wind at the dawnside downstream of the other two spacecraft. As shown in panel 1 of figure 3, TH-B observed an abrupt turning of the IMF from duskward to northward at UT 22:32, corresponding to UT 22:22, with a time lag of 10 minutes ((10+32.7) Re / (450 km/s)) from the subsolar magnetopause to TH-B. Periodical fluctuations were observed in both the TH-A and TH-E observations (figure 2), from ion density in panel 1, temperature in panel 2, magnetic field in panel 3 and 7, to velocity in panel 4 and 8, especially the alternating appearances of hot and cold ions in the energy-time

spectra (panel 5 and 9). The period was approximately 2 minutes (17 peaks within 34 minutes), and the tailward bulk propagation speed was approximately 212 km/s (3 Re / 90 s). In figure 3, the rotational characteristics were identified in the periodical fluctuations in $V_l$, $V_m$ and $V_n$ with phase differences between them. The magnetic field deviations in panels 3 and 5 indicated the perturbations of the magnetic field along with the deformation of the magnetopause. The alternating appearances of the two different plasmas imply the multiple periodic encounters of the magnetopause and the LLBL, which is one of the typical characteristics of K-H vortices.

In this event, the IMF is strongly northward, and the observed magnetic field doesn't change much, so it could be difficult to identify the magnetopause. We selected the four intervals of UT 22:24:00-22:24:40, UT 22:32:40-22:33:10, UT 22:35:50-22:36:10, UT 22:28:50-22:39:20, marked by the black arrows, when the TH-A ion spectrum showed the magnetosheath feature. During the four intervals, TH-A observed magnetosheath cold ions without magnetospheric hot ions (green regions at top of panel 5, figure 2). The absence of hot ions indicated that the spacecraft had crossed the magnetopause into the magnetosheath, where the outbound and inbound crossings of the magnetopause can be identified in the ion spectrum. At each pair of traversals, the local magnetopause coordinates LMN were calculated by using MVA (Sonnerup & Cahill, 1967; 1968). The details and results of MVA calculations are listed in table 1. In the calculations of MVA, relative large ratios of the second to third eigenvalues $r_{23}$ =$\varepsilon_2/\varepsilon_3$ means better reliability of determination of local coordinates. In the MVA results, it can be seen that 4 of 8 eigenvalue ratios are larger than 3, indicating the good reliability of the MVA method at their corresponding crossings, even though the magnetic field doesn't change strongly. At least at these traversals, the magnetopause was deformed into the nonlinear vortices. In some previous research, the threshold of the eigenvalue ratio was taken as 4 (e.g. Sergeev et. al., 2006). As for our results, at least, the eigenvalue ratios at the first pair of traversals are larger than 4, which mean that the calculated LMN coordinates at the outbound and inbound of the magnetopause are reliable and the magnetopause was deformed into a vortex. The calculated normal direction *N* as well as the tangential direction *M* of the local magnetopause is used to identify the distorted magnetopause. In each panel of figure 4, the normal and tangential directions *M-N* at the outbound and inbound magnetopause are plotted in the equatorial plane, compared with the average *M-N* of the magnetopause. The average magnetopause in dotted line, as well as the average *M-N* directions, is calculated from the model (Shue, 1998), and the dotted line is also approximately the trajectory of the spacecraft TH-A, which is moving at a relatively slow speed of about 2 km/s at the apogee. The distorted magnetopause is plotted in black line, perpendicular to *N* and parallel to *M* at outbound and inbound. The deviations of the *M-N* directions from the averaged magnetopause illustrate the magnetopause distortions formed by the K-H vortices. Such distortions of the magnetopause qualitatively explain the periodically alternating encounters of magnetosphere-like and magnetosheath-like plasmas. The plasma rotation is also illustrated by the red circle with arrow, consistent with the observations in panel 4 of figure 2.

The high-speed and low-density feature is one of the fundamental characteristics of rolled-up vortices (Nakamura et al., 2004; Takagi et al., 2006), and has been used to identify vortices in a single spacecraft measurements (e.g., Hasegawa et al., 2006; Hwang et. al., 2011, Grygorov et al., 2016). We estimated the magnetosheath velocity by averaging the TH-A measurements during the four magnetosheath intervals mentioned above, with the magnetosheath velocity of about 134

km/s. Figure 5 shows the $V_m$-$N_i$ plot, in which the blue lines mark the high-speed and low-density
region. $V_m$ is the tailward velocity, the $M$ component of the measured velocity expressed in the
averaged magnetopause coordinates LMN. Substantial data points are distributed in blue box in
figure 5, and the high-speed low-density feature can be seen in the Ni-Vm plot. Hence then,
although the surface waves can also explain some of the observations, the rotations of the plasma
flows, the perturbations of the magnetic field, the high-velocity and low-density feature, and the
distortions of the magnetopause support the likely formation of rolled-up K-H vortices. However,
the low eigenvalue ratios at some traversals of the magnetopause and the uncertainty of estimating
the magnetosheath velocity would admittedly degrade the evidence for the K-H vortices. It is
worth noting that the magnetopause oscillations started as soon as the IMF turned northward at
UT 22:22, which can facilitate the K-H instability, or else, the surface waves were amplified by
the K-H instability.
Before and after the UT 22:22-22:52 interval, the magnetospheric hot ions dominated in panel 5
of figure 2, mainly in the 3-25 keV range with an energy flux of $10^6$ eV/(cm$^2$-s-sr-eV), and the
magnetospheric hot electrons dominated in panel 6, mainly in the 0.5-25 keV range with an
energy flux of over $10^7$ eV/(cm$^2$-s-sr-eV). The typical temperatures of magnetospheric hot ions
and electrons were about 4 keV and 0.3 keV, respectively. On the other hand, during the UT
22:22-22:52 interval, the repeating magnetosheath cold ions in panel 5 were primarily observed
between 0.1-3 keV with an energy flux of over $10^6$ eV/(cm$^2$-s-sr-eV), and the cold electrons in
panel 6 were observed between 10-500 eV, with an energy flux of over $10^7$ eV/(cm$^2$-s-sr-eV). The
typical temperatures of magnetosheath cold ions and electrons were about 0.2 keV and 0.05 keV,
respectively. Embedded in the plasmas of the two different origins, the coexisting hot and cold
ions overlapped. Taking the mass ratio of protons to electrons into account, the gyro-radius of the
electrons is only 1/42 of protons with the same energy and the same magnetic field, estimated to
be approximately 2 km. We understand the ion transport as the coexistence of magnetosheath and
magnetospheric ions in the observations, characterized by the substantial cold ions in the steady
background of the hot plasma. For the proton's gyro-radius of approximately 80-100 km at the
magnetopause, the coexistence of the hot and cold ions in the spectrum is not sufficient to
diagnose the mixture of the two components. Thus, we used the observed hot electrons as an
additional indicator of the magnetosphere region because of their relatively smaller gyro-radius.
Hence, the criteria to identify the mixture/transport are described such that the cold ions of 0.1-3
keV can be observed with an energy flux over $10^5$ eV /(cm$^2$-s-sr-eV) in the hot ions background,
with an energy flux over $10^6$ eV /(cm$^2$-s-sr-eV), as well as a substantial enhancement in the energy
flux of the hot electrons of 0.5-5 keV. Based on such criteria, the ion mixture/transport intervals
were diagnosed from both TH-A and TH-E, marked by the green bars at the bottom of panel 6 and
the black bars at the bottom of panel 10 in figure 2. The transport regions in the TH-A
observations (green bars) were distributed at the edges of the vortices and appeared to be more
periodic, while those in the TH-E observations (black bars) were more dispersive. Such an
evolution implies the possible plasma transport, although a pre-existing LLBL or the difference of
spacecraft's distances to the magnetopause can also be a potential source.
The coexistence of hot and cold ions is one direct feature of the solar wind transport into
the magnetosphere, as clearly displayed in Geotail observations by Fujimoto et. al. (1998) and
in Cluster observations by Hasegawa et. al. (2004). In this event, the coexistence of hot and
cold ions was firstly noted near the periodically oscillating magnetopause. Furthermore, we

used the enhancement of hot electron flux as an indicator of the magnetosphere, and set up the more critical criteria to diagnose the coexistence, and hence to display the transport regions, as marked by the green bars at the bottom of panel 6 and black bars at the bottom of panel 10 in figure 2. By comparing the green bars and the black bars, it can be found that the transport regions in TH-A observations appears more periodic but those in TH-E observations more dispersed. The difference between the features of transport regions at upstream TH-A and downstream TH-E implies the plasma transport significantly occurred and evolved during the tailward propagation, along with the collapse of the vortices, leading to a kind of turbulence state, as illustrated in previous simulations (Nakamura et al., 2004; Matsumoto & Hoshino, 2004).

Intuitively, TH-E might be located more inner in the LLBL than TH-A, and observed more dispersive oscillations. TH-A observed very clearly periodic motions of magnetopause during the 34 minutes except UT 22:46-22:50, TH-E observed relatively much more dispersed spectrum during the interval but 5 clear oscillations appeared again during UT 22:40-22:48. However, it seems true that, on the whole, the spectrum observed at TH-E is much more turbulent than the periodic spectrum at TH-A. Such an evolution implies the collapse of the vortices and the evolution leading to turbulence state. In previous simulations (Nakamura et al., 2004; Matsumoto & Hoshino, 2004), the vortices collapse and cause transport of the solar wind into magnetosphere, after that, new vortices may be generated at the recovered magnetopause. The 5 oscillations during UT 22:40-22:48 at downstream TH-E can by explained as newly formed vortices. As mentioned above, the first K-H wave, as well as the transport regions arrived at the upstream TH-A as soon as the IMF abruptly turned northward. The K-H vortices were evidently activated as a response to the abrupt northward turning of the IMF, which was the direct change to facilitate the K-H instability immediately.

Previously, both electron and ion distributions were used to diagnose the region of observation (Chen et al., 1993). While diagnosing the transport regions in this event, the typical plasma features in different regions were selected for comparisons (figure 6), as illustrated by the energy flux distributions of both ions (blue line) and electrons (red line). In panels 1, both the ion and electron fluxes show single-peak at the low energy, indicating the components of cold and dense magnetosheath plasma. In panel 2, the ion flux shows a double-peak, which means the coexistence of the magnetosheath cold ions and magnetospheric hot ions. The relatively smaller peak/enhancement in the electron flux show that the magnetospheric hot electrons are detected, but the cold electrons dominate, implying the spacecraft is located in magnetosheath but very close to the magnetospause, a coexistence region. In panel 3, both the ion and electron fluxes show double-peak. The double-peak of the ion flux indicates coexistence of the magnetosheath cold ions and magnetospheric hot ions. For the electron flux, the peak at the high energy indicates that more magnetospheric hot electrons are detected, implying that the spacecraft is located in magnetosphere, another example of coexistence region. In panel 4, both ion and electron fluxes show single-peak at the high energy, indicating the components of hot and tenuous magnetospheric plasma. It should be noted that the ion flux plots (blue lines in each panel) should be lower in the tail, but show no such decrease tails in part because the data were absent at the high energy channels. The typical regions shown correspond to the magnetosheah, the energetic particle streaming layer, the LLBL, and the magnetosphere (Sibeck, 1991).

## 4 Summary

We analyzed observations from TH-A and TH-E that periodically encountered the magnetopause and the LLBL. Although they could be possibly caused by surface waves, the periodical encounters, characterized by the rotation features in the bulk velocity, magnetic field deviations, the high-speed low-density features and the distortions of the magnetopause deduced by MVA, showed the likely generation of K-H vortices. The K-H vortices started, or else, the surface waves were amplified by the K-H instability as soon as the IMF turned northward abruptly, which is the direct change to facilitate the instability immediately. By considering the enhancement of the hot electrons as an indicator of the magnetosphere region, typical plasma features were observed in different regions such as the energetic particle streaming layer, the LLBL, and the magnetosphere. The evolution between periodic and dispersed magnetopause observations from TH-A to TH-E implied the possible plasma transport, which is consistent with the different features of the coexisting regions of cold and hot plasmas between TH-A and TH-E. These new observations can complement existing observations and enhance our understanding of the plasma transport processes in K-H vortices.

**Data Availability**

The data for this paper are available at the Coordinated Data Analysis Web of NASA's Goddard Flight Center (http://cdaweb.gsfc.nasa.gov/istp_public/).

**Authors Contribution**

G. Q. Y. designed the idea and carried out the investigations, prepared the manuscript with contributions from all co-authors. G. K. P, C. L. C, and T. C. offered the valuable scientific discussions and helped to improve the manuscript. J. P. M. ensured the data and gave valuable suggestions. R. Y. prepared some of the figures.

**Competing Interests**

The Authors declare that they have no conflict of interest.

**Acknowledgements**

This work was supported by the Strategic Pioneer Program on Space Science, Chinese Academy of Sciences, Grant No. XDA15052500, XDA15350201, and XDA17010301, and by the National Natural Science Foundation of China, Grant No. 41574161, 41731070, 41574159 and 41004074. The authors are grateful to NASA's Goddard Flight Center and the associated instrument teams for supplying the data. The Authors thank Professor Chi Wang and Professor Lei Dai for valuable scientific discussions. The authors also express their thanks for the support from the Specialized Research Fund for State Key Laboratories and the CAS-NSSC-135 project. Part of the work was done during G. Q. Yan's visit at UC Berkeley, who cordially appreciates the assistance from Professor Forrest S. Mozer.

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

Figures and Captions

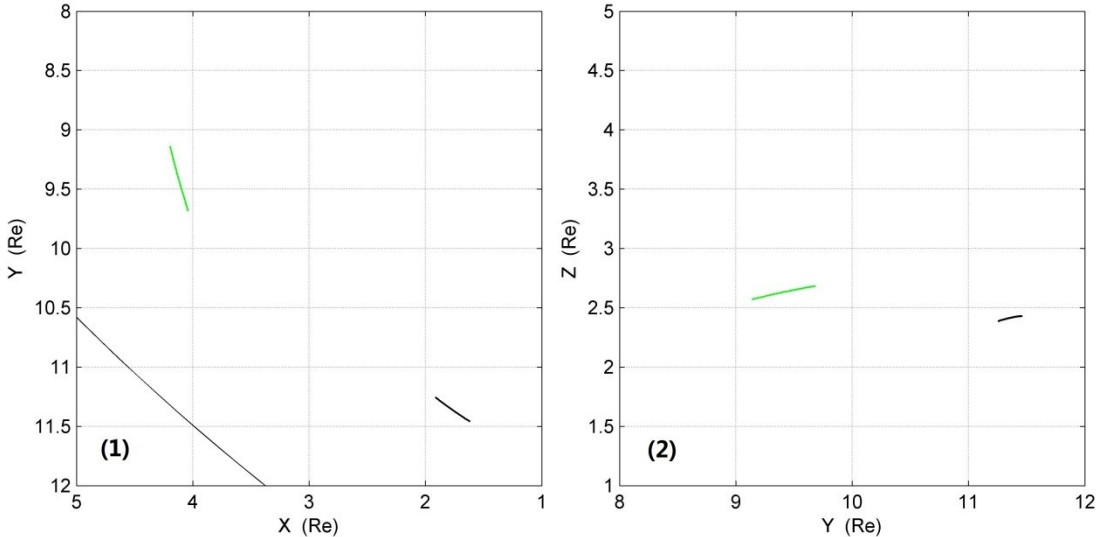

Figure 1. The orbits and positions of TH-A (green) and TH-E (black) during the interval of interest UT 22:20 ~ UT
22:54. The position data are expressed in GSM coordinates.















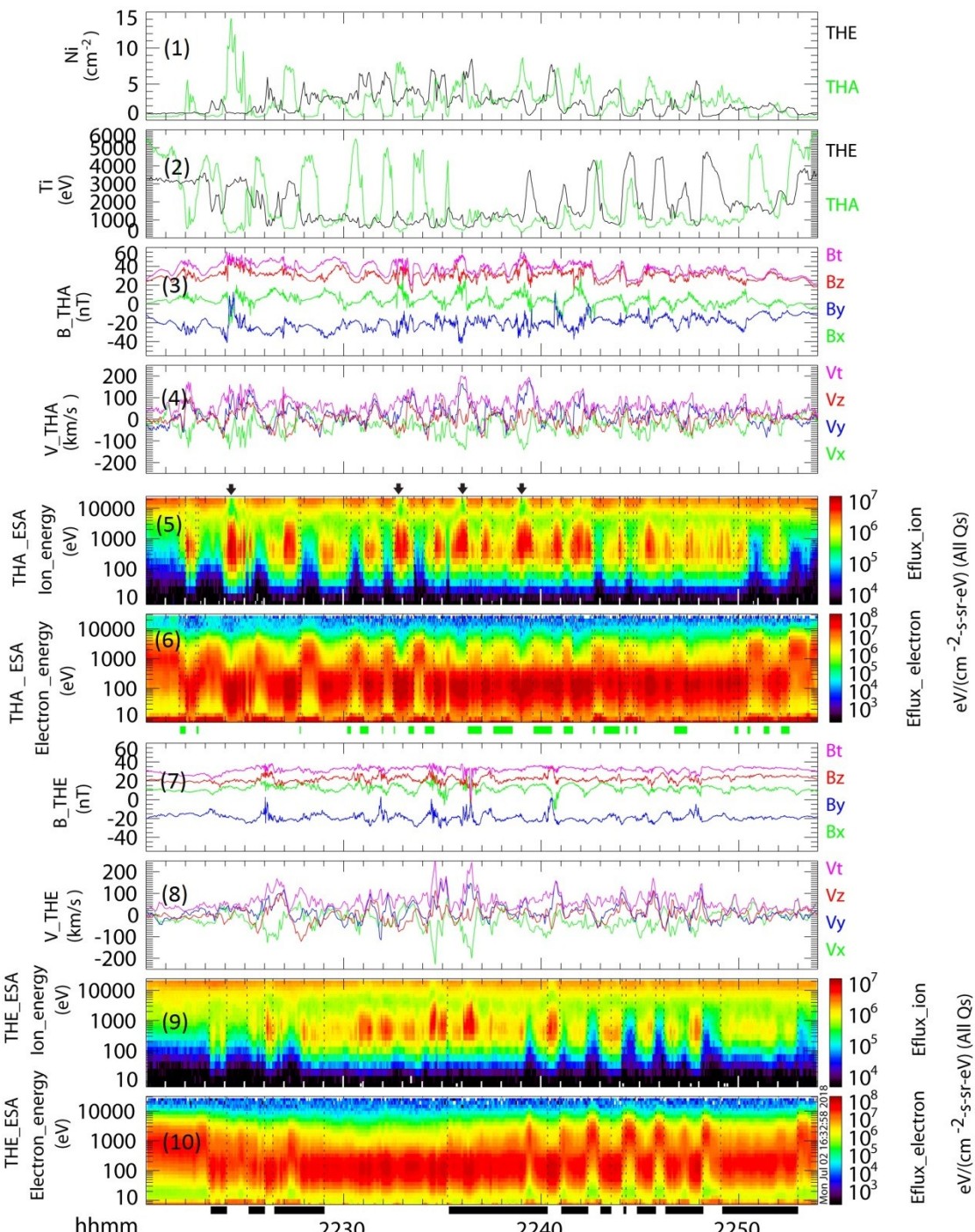

Figure 2. Fluctuations in the plasma parameters and the ion and electron energy-time spectra. Panel 1 is the ion densities from TH-A as a green line and from TH-E as a black line; panel 2 is the ion temperatures from TH-A as a green line and from TH-E as a black line; panels 3 and 4 are the magnetic field vectors and the ion bulk velocity vectors from TH-A, respectively; panels 5 and 6 are the ion and electron energy-time spectra from TH-A, respectively; panels 7 and 8 are the magnetic field vectors and the ion bulk velocity vectors from TH-E, respectively; panels 9 and 10 are the ion and electron energy-time spectra from TH-E, respectively. Vectors are all expressed in GSM coordinates. The four black arrows mark on top of panel 5 the TH-A intervals in the magnetosheath. The green bars on the bottom of panel 5 and the black bars on the bottom of panel 9 mark the transport regions in TH-A and TH-E observations, respectively, identified based on the criteria dictated in the text.

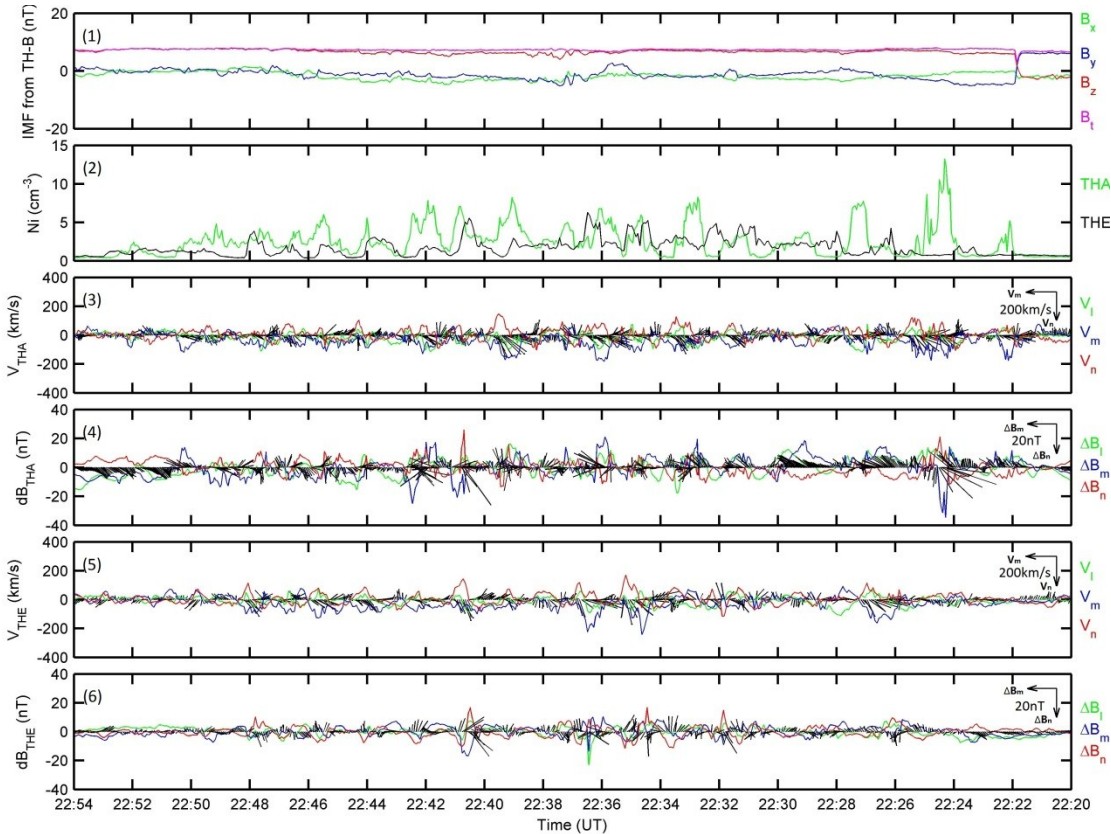

Figure 3. The observed plasma rotations and perturbations of the magnetic field because of the formation of K-H vortices. Panel 1 is the IMF monitored by TH-B near the lunar orbit, with a time lag of 10 minutes from subsolar magnetopause to TH-B; panel 2 are the ion densities from TH-A in green and from TH-E in black; panels 3 and 5 are the ion bulk velocities from TH-A and TH-E, respectively, expressed in averaged local magnetopause coordinates LMN, deduced from the magnetopause model (Shue et al., 1998); panels 4 and 6 are the magnetic field perturbations, $\Delta\boldsymbol{B}=\boldsymbol{B}-\boldsymbol{B}_{\mathrm{mean}}$, from TH-A and TH-E respectively, expressed in LMN. Note that the time begins from the right and passes to the left, so that the $M$ component orients leftward and N component orients downward in the plots.

Table 1. Results of MVA analysis at the four magnetosheath encounters of TH-A. The ratio of the second to third
eigenvalues $r_{23}=\varepsilon_2/\varepsilon_3$ are shown in the right column.

| Num | Time interval | L | M | N | $r_{23}=\varepsilon_2/\varepsilon_3$ |
|---|---|---|---|---|---|
| 1 | 22:23:50-22:24:12 | 0.0637 | 0.4374 | 0.8970 | 4.56 |
| | | -0.3955 | -0.8141 | 0.4251 | |
| | | 0.9162 | -0.3819 | 0.1212 | |
| 2 | 22:24:20-22:25:15 | 0.0646 | -0.3877 | -0.9195 | 5.27 |
| | | -0.2602 | 0.8830 | -0.3906 | |
| | | 0.9634 | 0.2645 | -0.0438 | |
| 3 | 22:32:30-22:32:52 | 0.0017 | 0.8349 | 0.5504 | 1.82 |
| | | -0.6860 | -0.3995 | 0.6081 | |
| | | 0.7276 | -0.3786 | 0.5721 | |
| 4 | 22:32:52-22:33:14 | -0.0561 | -0.3946 | -0.9171 | 2.25 |
| | | 0.3303 | 0.8595 | -0.3900 | |
| | | 0.9422 | -0.3248 | 0.0821 | |
| 5 | 22:35:35-22:36:00 | 0.2636 | 0.1004 | 0.9594 | 3.34 |
| | | -0.4912 | -0.8420 | 0.2231 | |
| | | 0.8302 | -0.5301 | -0.1726 | |
| 6 | 22:36:07-22:36:20 | -0.0102 | 0.3363 | -0.9417 | 2.77 |
| | | 0.0117 | 0.9417 | 0.3362 | |
| | | 0.9999 | -0.0076 | -0.0135 | |
| 7 | 22:38:41-22:39:05 | 0.2307 | 0.0363 | 0.9724 | 3.42 |
| | | -0.4125 | -0.9014 | 0.1315 | |
| | | 0.8813 | -0.4314 | -0.1930 | |
| 8 | 22:39:05-22:40:30 | -0.0574 | -0.5073 | -0.8599 | 1.07 |
| | | -0.7802 | -0.5145 | 0.3556 | |
| | | 0.6229 | -0.6913 | 0.3662 | |



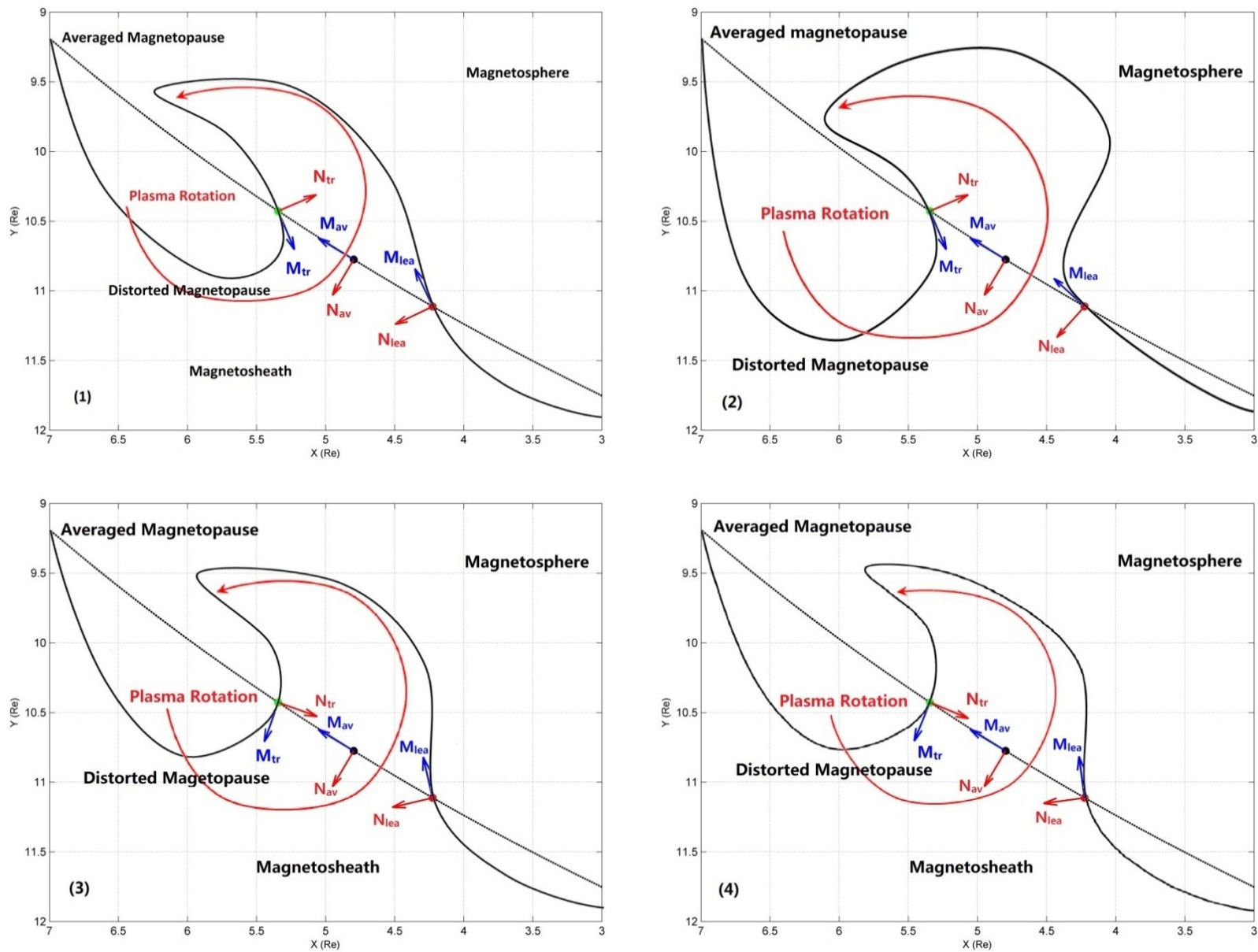

Figure 4. The magnetopause distortions formed by the K-H vortices deduced by the MVA. The average magnetopause (dashed lines), approximated to the spacecraft trajectory, was calculated from the magnetopause model (Shue et al., 1998). Traversal pair at UT 22:24 in panel 1: $M_{lea}$=(0.4374, -0.8141, -0.3819) and $N_{lea}$=(0.8970, 0.4251, 0.1212) at the outbound crossing; $M_{tr}$=(-0.3877, 0.8830, 0.2645) and $N_{tr}$=(-0.9195, -0.3906, -0.0438) at the inbound crossing. Traversal pair at UT 22:32 in panel 2: $M_{lea}$=(0.8349, -0.3995, -0.3786) and $N_{lea}$=(0.5504, 0.6081, 0.5721) at the outbound crossing; $M_{tr}$=(-0.3946, 0.8595, -0.3248) and $N_{tr}$=(-0.9171, -0.3900, 0.0821) at the inbound crossing. Traversal pair at UT 22:36 in panel 3: $M_{lea}$=(0.1004, -0.8420, -0.5301) and $N_{lea}$=(0.9594, 0.2231, -0.1726) at the outbound crossing; $M_{tr}$=(0.3363, 0.9417, -0.0076) and $N_{tr}$=(-0.9417, 0.3362, -0.0135) at the inbound crossing. Traversal pair at UT 22:39 in panel 4: $M_{lea}$=(0.0363, -0.9014, -0.4314) and $N_{lea}$=(0.9724, 0.1315, -0.1930) at the outbound crossing; $M_{tr}$=(-0.5073, -0.5145, -0.6913) and $N_{tr}$=(-0.8599, 0.3556, 0.3662) at the inbound crossing.

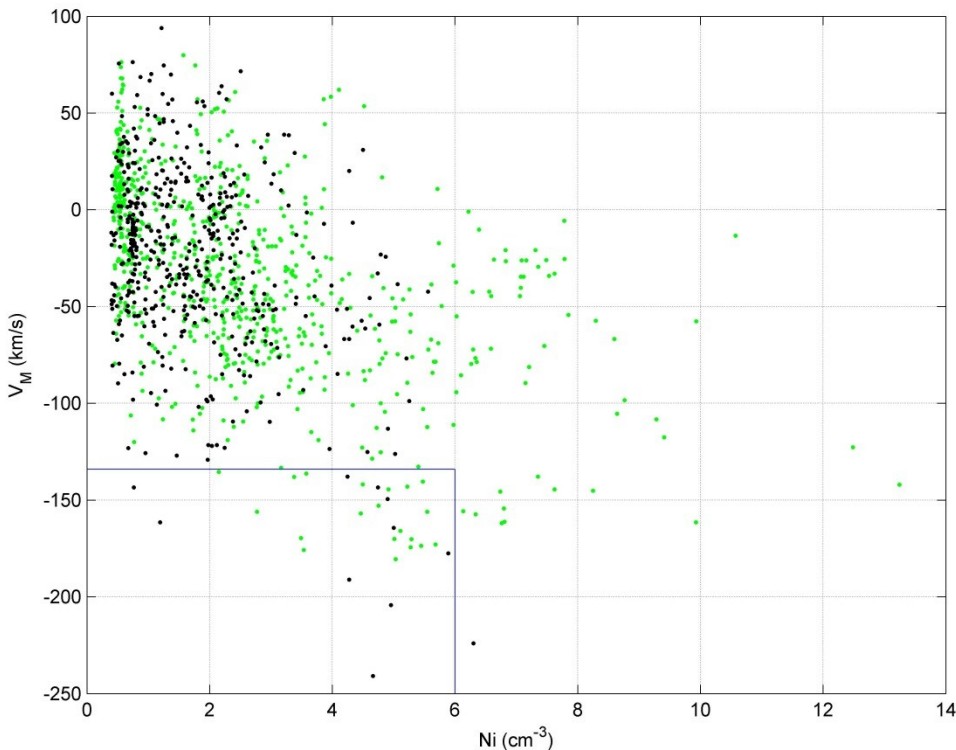

Figure 5. The observed velocity along the tailward direction versus the ion density. Green dots are from TH-A

observations and black dots from TH-E observations. The blue lines mark the high-speed and low density region

possibly caused by the acceleration of the rotation.

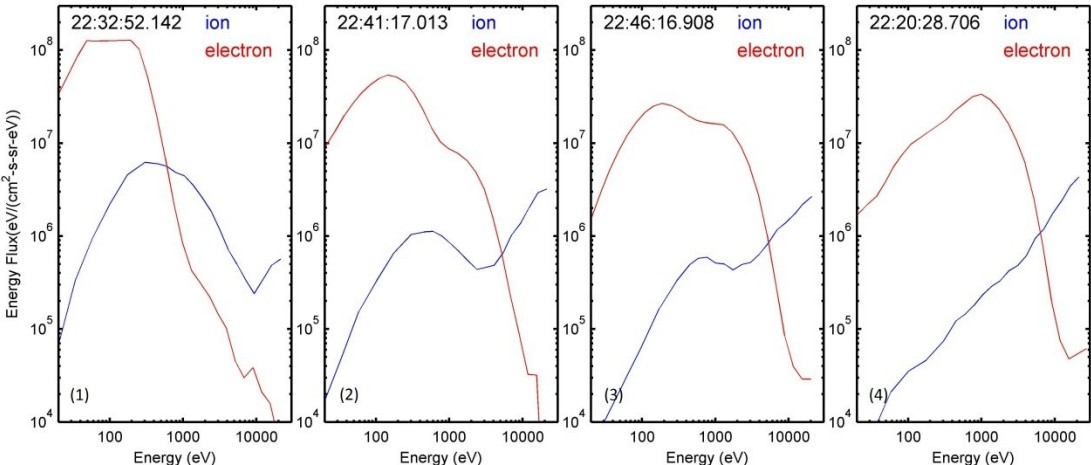

Figure 6. Typical portraits of the energy-time spectra of plasmas in different regions. Panel 1 is the magnetosheath observed by TH-A at 22:32:52.142; panel 2 is co-existence region I observed by TH-A at 22:41:17.013; panel 3 is co-existence region II observed by TH-E at 22:46:16.908; panel 4 is the magnetosphere observed by TH-A at 22:20:28.706.