# Peer review of "Plasma transport into the duskside magnetopause caused by"

_Annales Geophysicae, 2019_

## Referee Comment (RC1) · Anonymous Referee #1 · 9 Sep 2019

The paper reports observations of Kelvin-Helmholtz (KH) vortices by two THEMIS spacecraft (THA and THE) at the dusk magnetopause, dayside of the terminator. The periodic crossings of the magnetopause occurred following a northward turning of the interplanetary magnetic field. The identification of the vortices is based on the computation of boundary normal directions via minimum variance analysis (MVA). Interestingly, low density plasma faster than magnetosheath plasma – a common feature of KH vortices – was not observed. The spacecraft locations allow for an assessment of

the evolution of the vortices: Crossing of regions with mixed magnetospheric and magnetosheath plasmas appear more/less periodic at an earlier/later stage, suggesting the transport of plasma across the magnetopause.

My main criticism is related to the identification of vortices and the interpretation of observations supporting the hypothesis of plasma transport across the magnetopause. At this point, I do not think that the conclusions of the paper are sufficiently supported by the observations.

Specific comments:

1) It is not convincingly shown that the magnetopause oscillations observed are actually due to the passage of magnetopause KH vortices and not, e.g., due to the passage of magnetopause surface waves that have not yet reached the non-linear stage. In this study, vortices are mainly identified by a sequence of boundary normal vectors, obtained from MVA applied to magnetic field observations (e.g., line 122+). However, MVA results can strongly depend on the selected time intervals around current sheets to which the method is applied. It would help enormously if the authors could assess the stability and reliability of the MVA results, also taking into account the eigenvalue ratios as described by Sergeev et al. (Ann. Geophys., 2006). I am doubtful about the reliability of MVA results here, because the magnetic field variations that can be analyzed are not particularly strong (see panels 3 and 7 of Figure 2).

Furthermore, I am concerned about the identification of KH vortices in the absence of low density plasma that is faster than the magnetosheath plasma (e.g., line 141+). This feature has been used to identify vortices using single spacecraft measurements. It may also be observed without vortices being present: the passage of a surface wave should suffice. But if vortices are present, then the feature should be observable too, and I cannot find any statement in Masson and Nykyri (2016) that would suggest the opposite. So the absence of that feature indicates, in my opinion, that the oscillations are rather related to a magnetopause surface wave rather than to KH vortices. Note that rotations in the bulk velocity, magnetic field deviations, and distortions of the magnetopause can also result from magnetopause surface waves (e.g., line 205).

2) It is not convincingly shown that a significant and unambiguous plasma transport took place across the magnetopause (e.g., lines 211, 214). This main conclusion of the paper is inferred from observation of less periodic features seen by THE in comparison to THA, the former being located further down the tail from the latter. I would at least expect some further discussion on how this strong conclusion can be drawn from the observations (e.g., by putting the results into the context of prior observations or simulations). However, also the observations themselves are not consistent over the presented time interval: As can be seen in Figure 2, THA sees more periodic magnetopause oscillations before 22:40 UT, as discussed in the paper. After 22:40, THE sees very periodic oscillations and THA observations are "more dispersed" (e.g., between 22:44 and 22:51). Following the argument in the manuscript, plasmas should have unmixed while vortices moved from THA to THE during this period of time.

3) It is not convincingly shown that there was no pre-existing (low-latitude) boundary layer (LLBL), consisting of a mixture of magnetosheath and magnetospheric plasmas. This mixture is used as a synonym to plasma transport across the magnetopause in the paper (line 164), supposedly starting with the northward turning of the IMF. But a LLBL might have been present at the magnetopause even before the oscillations started. To confirm this or rule it out, we would need spacecraft observations across the magnetopause near the THEMIS positions before the event. However, such observations do not seem to be available, and both THA and THE were probably too far away from the magnetopause to observe a pre-existing LLBL. During the event, as the surface waves went by, the magnetopause moved periodically closer to the spacecraft so that they were able to enter the LLBL, as stated in line 204.

4) I do not know what the authors exactly mean by "field-line stretching" (e.g., lines 22, 118) and how such a behavior would be reflected or identifiable in single spacecraft ion velocity or magnetic field time series.

СЗ

Minor comments on figures:

- Figure 1 conveys very little information. It may be sufficient to keep only the x-y-plot.

- It may be helpful for the reader to state the meaning of the green and black bars in Figure 2 (below panels 6 and 10) in the caption.

- I cannot see any reason for the inverted time line in Figure 3. Please state a clear reason or display the data in the conventional way, with time moving forward to the right.

---

## Referee Comment (RC2) · Anonymous Referee #2 · 17 Oct 2019

In this research, the authors have investigated the transport of the solar wind plasmas into the magnetosphere from the flank side during the northern IMF period, and present a clear evidence for the K-H instability mechanism. The THEMIS data are used and MVA method is applied to determine the configuration of the distorted magnetopause. The periodic K-H vertices are observed and there exists the mixture of the cold magnetosheath plasmas and hot magnetospheric plasmas within the vertices. This paper will enhance the understanding on the mechanism how the K-H instability drives the

transport of the magnetosheath plasmas into the magnetotail plasma sheet. So it can be accepted for publication after minor modifications. Some comments on the paper are as the following. (1) Line 127-129: L and M are tangential to the magnetopause, so the word "parallel" can be replaced by "tangential". In line 129, the expression "the thangential and normal directions M-N" is proper. (2) Line 152-159: The values of the temperatures of the hot magnetospheric plasmas and cold magnetosheath ions and electrons can be given. (3) Line 347: The topic of the paper is not complete.

---

## Author Comment (AC1) · 25 Oct 2019

Thanks you very much for your spending time evaluating our article. Your comments and critics help us to improve our understanding of the observations in this event and to find more accurate descriptions. Based on our thinking over some questions evolved in this event, we would like to response as follows. Your original comments and questions are in blue and our responses are in square brackets.

[Figure]

The paper reports observations of Kelvin-Helmholtz (KH) vortices by two THEMIS spacecraft (THA and THE) at the dusk magnetopause, dayside of the terminator. The periodic crossings of the magnetopause occurred following a northward turning of the interplanetary magnetic field. The identification of the vortices is based on the computation of boundary normal directions via minimum variance analysis (MVA). Interestingly, low density plasma faster than magnetosheath plasma – a common feature of KH vortices – was not observed. The spacecraft locations allow for an assessment of the evolution of the vortices: Crossing of regions with mixed magnetospheric and magnetosheath plasmas appear more/less periodic at an earlier/later stage, suggesting the transport of plasma across the magnetopause. My main criticism is related to the identification of vortices and the interpretation of observations supporting the hypothesis of plasma transport across the magnetopause. At this point, I do not think that the conclusions of the paper are sufficiently supported by the observations. Specific comments: 1) It is not convincingly shown that the magnetopause oscillations observed are actually due to the passage of magnetopause KH vortices and not, e.g., due to the passage of magnetopause surface waves that have not yet reached the non-linear stage. In this study, vortices are mainly identified by a sequence of boundary normal vectors, obtained from MVA applied to magnetic field observations (e.g., line 122+). However, MVA results can strongly depend on the selected time intervals around current sheets to which the method is applied. It would help enormously if the authors could assess the stability and reliability of the MVA results, also taking into account the eigenvalue ratios as described by Sergeev et al. (Ann. Geophys., 2006). I am doubtful about the reliability of MVA results here, because the magnetic field variations that can be analyzed are not particularly strong (see panels 3 and 7 of Figure 2).

[ You are right that the MVA results strongly depend on the selected time intervals that cover the magnetopause. In this event, the IMF is strongly northward, and the observed magnetic field doesn't change much, so it could be difficult to identify the magnetopause. So, we selected the four intervals when the TH-A ion spectrum shows the magnetosheath feature (absence of the magnetospheric hot ions) to calculate the

local boundary coordinates. We operated the MVA method carefully by using the high resolution magnetic field data, and by identifying the magnetopause in the ion spectrum. In diagnosing the MVA results, we also selected the better results with relative larger ratios between the eigenvalues. As you suggested above, we would like to show the details of the MVA analysis in a table, which will also be added to the revision of our article. In the MVA results ( in another pdf as the supplement), it can be seen that 4 of 8 eigenvalue ratios are larger than 3, indicating the reliability of the MVA method at their corresponding crossings, even though the magnetic field doesn't change strongly. At least at these traversals, the magnetopause was deformed to the nonlinear vortices. ] Furthermore, I am concerned about the identification of KH vortices in the absence of low density plasma that is faster than the magnetosheath plasma (e.g., line 141+). This feature has been used to identify vortices using single spacecraft measurements. It may also be observed without vortices being present: the passage of a surface wave should suffice. But if vortices are present, then the feature should be observable too, and I cannot find any statement in Masson and Nykyri (2016) that would suggest the opposite. So the absence of that feature indicates, in my opinion, that the oscillations are rather related to a magnetopause surface wave rather than to KH vortices. Note that rotations in the bulk velocity, magnetic field deviations, and distortions of the magnetopause can also result from magnetopause surface waves (e.g., line 205).

[ We agree that the high-speed low-density feature is one of the typical characteristics of the K-H vortices and very useful in diagnosing the K-H vortices in single spacecraft measurements. It was a surprise to us that the high-speed low-density feature did not appear in the Ni-Vm plot in this event. We used to estimate the magnetosheath velocity by drawing a horizontal line that is close to most of the magnetosheath intervals shown in panel 4 of figure 2. The horizontal line was at the velocity of about 180 km/s. Now, we re-estimate the magnetosheath velocity by averaging the TH-A measurements during the four magnetosheath intervals, with the more accurate velocity of about 134 km/s in the magnetosheath. Based on the new estimation, the high-speed low-density feature can be seen in the Ni-Vm plot, with more data points distributed in blue box

(see the revised figure 5 below). The high-speed low-density feature can support the K-H vortices in this event. Furthermore, the linear surface waves could not explain the fine structure of the observed first perturbation in TH-E observations, show in the additional figure below, which we are further working on, with new results of the microphysical process to transport solar wind into magnetosphere within the K-H vortices in another article. The double peaks in the ion measurements can be caused either by second traversal of the non-linear K-H vortex or by the secondary substructure of the vortex. More details of the plasma transport in the K-H vortex will be revealed in another article in preparation.] 2) It is not convincingly shown that a significant and unambiguous plasma transport took place across the magnetopause (e.g., lines 211, 214). This main conclusion of the paper is inferred from observation of less periodic features seen by THE in comparison to THA, the former being located further down the tail from the latter. I would at least expect some further discussion on how this strong conclusion can be drawn from the observations (e.g., by putting the results into the context of prior observations or simulations). However, also the observations themselves are not consistent over the presented time interval: As can be seen in Figure 2, THA sees more periodic magnetopause oscillations before 22:40 UT, as discussed in the paper. After 22:40, THE sees very periodic oscillations and THA observations are "more dispersed" (e.g., between 22:44 and 22:51). Following the argument in the manuscript, plasmas should have unmixed while vortices moved from THA to THE during this period of time. [ Thank you for your comments that can further our thinking over some questions in this event. Above all, word "unambiguous" in conclusion has been deleted so that it is not so strong. The coexistence of hot and cold ions is one of direct feature of the solar wind transport into magnetosphere, as clearly displayed in Geotail observations by Fujimoto et. al. (1998) and in Cluster observations by Hasegawa et. al. (2004). In this event, the coexistence of hot and cold ions was firstly noted near the periodically oscillating magnetopause. Furthermore, we used the enhancement of hot electron flux as the indicator of the LLBL, and set up the more critical criteria to diagnose the coexistence, and hence to display the transport regions, as marked by

the green bars at the bottom of panel 6 and black bars at the bottom of panel 10 in figure 2. Compared with the possible pre-existing LLBL before the perturbations, the coexistence of hot and cold ions shows the fresh entering of cold ions into the LLBL. The evidence of the plasma transport is clearly shown in this event. By comparing the green bars and the black bars, it can be found that the transport regions in TH-A observations appears more periodic but those in TH-E observations more dispersed. The difference between the features of transport regions at upstream TH-A and down-stream TH-E implies the plasma transport significantly occurred and evolved during the tailward propagation, along with the collapse of the vortices, leading to a kind of turbulence state, as illustrated in previous simulations (Nakamura et al., 2004; Matsumoto & Hoshino, 2004). You are right that TH-A observed very clearly periodic motions of magnetopause during the 34 minutes except UT 22:46-22:50 TH-E observed relatively much more dispersed spectrum during the interval but 5 clear periods of oscillations appeared again during UT 22:40-22:48. However, it seems true that on the whole the spectrum observed at TH-E is much more turbulent than the periodic spectrum at TH-A. Such characteristics imply the collapse of the vortices and the evolution leading to turbulence state. In previous simulations (Nakamura et al., 2004; Matsumoto & Hoshino, 2004), the vortices collapse and transport solar wind into magnetosphere, after that, new vortices may be generated at the recovered magnetopause. The 5 oscillations during UT 22:40-22:48 at downstream TH-E can by explained as newly formed vortices. If you agree with the above discussion, it will be added to the text to enrich the understanding of observations in this event. Thank you for all your comments, critics and suggestions that help us to improve the article.] 3) It is not convincingly shown that there was no pre-existing (low-latitude) boundary layer (LLBL), consisting of a mixture of magnetosheath and magnetospheric plasmas. This mixture is used as a synonym to plasma transport across the magnetopause in the paper (line 164), supposedly starting with the northward turning of the IMF. But a LLBL might have been present at the magnetopause even before the oscillations started. To confirm this or rule it out, we would need spacecraft observations across the magnetopause near the

THEMIS positions before the event. However, such observations do not seem to be available, and both THA and THE were probably too far away from the magnetopause to observe a pre-existing LLBL. During the event, as the surface waves went by, the magnetopause moved periodically closer to the spacecraft so that they were able to enter the LLBL, as stated in line 204. [ Thank you for your comments that have pushed us to think over the question further. It is true that we need observations of another spacecraft nearby across the magnetopause to confirm or rule out the possible pre-existing denser layer. We used to trying the MMS conjunctions but the four-spacecraft stellar were not located near THEMIS. From the observations in this event, we only know that neither of the 2 spacecraft of THEMIS near the magnetopause observed the pre-existing denser layer before the K-H waves (surface waves according to your comments). As you pointed out, "a LLBL might have been present at the magnetopause even before the oscillations started". To tell the truth, we cannot confirm the absence of the pre-existing denser layer before the perturbations. So the description that "there is no pre-existing denser layer to facilitate the instability" has been deleted. Thank you for your reminding that the word "mixture" was inappropriately used as a synonym to plasma transport across the magnetopause. Mixture is a state of two components in plasma, such as the plasma in LLBL, while transport is a process of the transfer of solar wind into the magnetosphere. The LLBL is a result or consequence of the solar wind transport into magnetosphere. In this event, the most prominent characteristics are the periodic oscillations of the magnetopause, and the coexistence of hot and cold ions, with more emphasis on the transport process. So we are using the word "transport" and "coexistence" to describe the event. The "mixture region" in the caption of figure 6 was also replaced by "coexistence region" to avoid misleading. ] 4) I do not know what the authors exactly mean by "field-line stretching" (e.g., lines 22, 118) and how such a behavior would be reflected or identifiable in single spacecraft ion velocity or magnetic field time series. [ Thank you for your suggestions. We used to describe the deformation of the magnetopause accompanied by the field line stretching as illustrated by Hasegawa et. al. (2004), the magnetic field was disturbed at the low latitude region.

Actually, the magnetopause deformation can cause the magnetic deviations, and the deviations can be available in both linear surface waves and nonlinear vortices, as you mentioned in your comments. In order to avoid misleading or misunderstanding, the "field line stretching" has been deleted for more accurate description. ]

Minor comments on figures: - Figure 1 conveys very little information. It may be sufficient to keep only the x-y-plot. [ Thanks for your suggestion. We revised the figure by taking only the X-Y plot indicating the location near the magnetopause and the Y-Z plot indicating the low-latitude region.] - It may be helpful for the reader to state the meaning of the green and black bars in Figure 2 (below panels 6 and 10) in the caption. [ Thank you for such a reminding. The description of the green and black bars has been added to the caption of figure 2.] - I cannot see any reason for the inverted time line in Figure 3. Please state a clear reason or display the data in the conventional way, with time moving forward to the right. [ In this event, it was at the duskside of magnetopause, we displayed the vectors of the velocity and magnetic field perturbations not only in plots but also in arrowed lines (black lines in panels 3-6), with the scales of the magnitudes on the right side of each panel. The directions of M and N components correspond to the leftward and downward directions respectively, viewing from the Z direction. The data on the right side occur earlier than those on the left side, earlier data should be propagated more tailward to the rightside. We used the reverted time line (as Hasegawa et al. (2004) used in their publication in Nature) just in order to show the time sequence from right to left. The illustrations of Vm, Vn, $\triangle$Bm, $\triangle$Bn have been added to the scales on the right side of panels 3-6.]

Please also note the supplement to this comment:
https://www.ann-geophys-discuss.net/angeo-2019-103/angeo-2019-103-AC1-supplement.pdf

───────────────────────

[Figure]

**Fig. 1.** Figure 5. The observed velocity along the tailward direction versus the ion density. Green dots are from TH-A observations and black dots from TH-E observations. The blue lines mark the high-speed and

[Figure]

**Fig. 2.** Additional figure, which is not enclosed in this article, to show the double peaks in TH-E plasma observations. The linear surface waves could not explain the double peaks.

[Figure]

**Fig. 3.** Figure 1. The orbits and positions of TH-A (green) and TH-E (black) during the interval of interest UT 22:20 ∼ UT 22:54. The position data are expressed in GSM coordinates.

---

## Author Comment (AC2) · 25 Oct 2019

Thank you very much for your spending your time evaluating our article. Your endorsement, as well as your suggestions, is encouraging us to go further to investigate more details of the transport mechanism in K-H vortices. We have made the minor modifications as you suggested, and would like to response as follows. Your original comments and questions are in blue and our responses are in aquare brackets.

[Figure]

In this research, the authors have investigated the transport of the solar wind plasmas into the magnetosphere from the flank side during the northern IMF period, and present a clear evidence for the K-H instability mechanism. The THEMIS data are used and MVA method is applied to determine the configuration of the distorted magnetopause. The periodic K-H vortices are observed and there exists the mixture of the cold magnetosheath plasmas and hot magnetospheric plasmas within the vortices. This paper will enhance the understanding on the mechanism how the K-H instability drives the transport of the magnetosheath plasmas into the magnetotail plasma sheet. So it can be accepted for publication after minor modifications. Some comments on the paper are as the following.

(1) Line 127-129: L and M are tangential to the magnetopause, so the word "parallel" can be replaced by "tangential". In line 129, the expression "the hangential and normal directions M-N" is proper. [Thank you for your valuable suggestion that helps us to describe the details more accurately. The text has been revised by replacing "parallel" by "tangential".] (2) Line 152-159: The values of the temperatures of the hot magnetospheric plasmas and cold magnetosheath ions and electrons can be given. [The typical temperatures have been given in the text.] (3) Line 347: The topic of the paper is not complete. [Sorry for the unexpected mistake. The topic of the paper has been complemented. ]

---

## Author Response (AR1)

**Dear Editors and Referees,**

We owe our great thanks to the editor and the referees for the evaluation of our article. The criticisms and suggestions have been helping us think over some questions in our investigation and improve our manuscript much.

According to the suggestions and comments of the referees, the manuscript has been revised:

1. The abstract and summary were rewritten so that it is in consistence with the text and conclusions, according to the suggestions of the referees.

2. Figure 1 was regenerated to remove the redundant plots according to one referee's suggestion, only the X-Y and Y-Z plots were remained.

3. Based on more accurate estimate of the magnetosheath velocity, the low-density and high velocity feature was confirmed, the  $V_m$ -Ni plot in figure 5 was regenerated with more data points in the low-density and high velocity region. Some further discussions were also added to the text correspondingly.

4. The results of the MVA were discussed more in the text, and a table of the results was added to the manuscript, to show the reliability of MVA in this event.

5. Per one referee's comments, further discussion was added to the text on the plasma transport, some descriptions were deleted to avoid confusing or misunderstanding.

6. Figure 3 was revised. The illustrations of  $V_m$ ,  $V_n$ ,  $\Delta B_m$ ,  $\Delta B_n$  were added to the scales on the right side of panels 3-6 to make the plots easier to read.

7. Some references were added to the reference list.

8. Some other mistakes were corrected in the text.

9. The Data availability, Author contribution and Competing interest were added to the manuscript as required by the system.

**Reply to Anonymous Referee #1**

Thank you very much for spending time to evaluate our article. Your comments and criticisms will help us to improve our understanding of the observations in this event and to find more accurate descriptions. Based on our thinking over some questions involved in this event, we would like to respond as follows: Your original comments and questions are in blue and our responses are in black.

The paper reports observations of Kelvin-Helmholtz (KH) vortices by two THEMIS spacecraft (THA and THE) at the dusk magnetopause, dayside of the terminator. The periodic crossings of the magnetopause occurred following a northward turning of the interplanetary magnetic field. The identification of the vortices is based on the computation of boundary normal directions via minimum variance analysis (MVA). Interestingly, low density plasma faster than magnetosheath plasma – a common feature of KH vortices – was not observed. The spacecraft locations allow for an assessment of C1 the evolution of the vortices: Crossing of regions with mixed magnetosheatic and magnetosheath plasma across the magnetopause.

My main criticism is related to the identification of vortices and the interpretation of observations supporting the hypothesis of plasma transport across the magnetopause. At this point, I do not think that the conclusions of the paper are sufficiently supported by the observations.

Specific comments:

1) It is not convincingly shown that the magnetopause oscillations observed are actually due to the passage of magnetopause KH vortices and not, e.g., due to the passage of magnetopause surface waves that have not yet reached the non-linear stage. In this study, vortices are mainly identified by a sequence of boundary normal vectors, obtained from MVA applied to magnetic field observations (e.g., line 122+). However, MVA results can strongly depend on the selected time intervals around current sheets to which the method is applied. It would help enormously if the authors could assess the stability and reliability of the MVA results, also taking into account the eigenvalue ratios as described by Sergeev et al. (Ann. Geophys., 2006). I am doubtful about the reliability of MVA results here, because the magnetic field variations that can be analyzed are not particularly strong (see panels 3 and 7 of Figure 2).

We think it right that the MVA results strongly depend on the selected time intervals that cover the magnetopause. In this event, the IMF is strongly northward, and the observed magnetic field doesn't change much, so it could be difficult to identify the magnetopause. So, we selected the four intervals when the TH-A ion spectrum shows the magnetosheath feature (absence of the magnetospheric hot ions) to calculate the local boundary coordinates. We operated the MVA method carefully by using the high resolution magnetic field data, and by identifying the magnetopause in the ion spectrum. In diagnosing the MVA results, we also selected the better results with relative larger ratios between the eigenvalues. As you suggested above, we would like to show the details of the MVA analysis in a table, which will also be added to the revision of our article. In the MVA results, it can be seen that 4 of 8 eigenvalue ratios are larger than 3, indicating the reliability of the MVA method at their corresponding crossings, even though the magnetic field doesn't change strongly. At least at these traversals, the magnetopause was deformed to the nonlinear vortices.

| Num | Time interval     | L       | М       | N       | $r_{23} = \epsilon_2 / \epsilon_3$ |
|-----|-------------------|---------|---------|---------|------------------------------------|
|     |                   | 0.0637  | 0.4374  | 0.8970  |                                    |
| 1   | 22:23:50-22:24:12 | -0.3955 | -0.8141 | 0.4251  | 4.56                               |
|     |                   | 0.9162  | -0.3819 | 0.1212  |                                    |
|     |                   | 0.0646  | -0.3877 | -0.9195 |                                    |
| 2   | 22:24:20-22:25:15 | -0.2602 | 0.8830  | -0.3906 | 5.27                               |
|     |                   | 0.9634  | 0.2645  | -0.0438 |                                    |
|     |                   | 0.0017  | 0.8349  | 0.5504  |                                    |
| 3   | 22:32:30-22:32:52 | -0.6860 | -0.3995 | 0.6081  | 1.82                               |
|     |                   | 0.7276  | -0.3786 | 0.5721  |                                    |
|     |                   | -0.0561 | -0.3946 | -0.9171 |                                    |
| 4   | 22:32:52-22:33:14 | 0.3303  | 0.8595  | -0.3900 | 2.25                               |
|     |                   | 0.9422  | -0.3248 | 0.0821  |                                    |
|     |                   | 0.2636  | 0.1004  | 0.9594  |                                    |
| 5   | 22:35:35-22:36:00 | -0.4912 | -0.8420 | 0.2231  | 3.34                               |
|     |                   | 0.8302  | -0.5301 | -0.1726 |                                    |
|     |                   | -0.0102 | 0.3363  | -0.9417 |                                    |
| 6   | 22:36:07-22:36:20 | 0.0117  | 0.9417  | 0.3362  | 2.77                               |
|     |                   | 0.9999  | -0.0076 | -0.0135 |                                    |
|     |                   | 0.2307  | 0.0363  | 0.9724  |                                    |
| 7   | 22:38:41-22:39:05 | -0.4125 | -0.9014 | 0.1315  | 3.42                               |
|     |                   | 0.8813  | -0.4314 | -0.1930 |                                    |
|     |                   | -0.0574 | -0.5073 | -0.8599 |                                    |
| 8   | 22:39:05-22:40:30 | -0.7802 | -0.5145 | 0.3556  | 1.07                               |
|     |                   | 0.6229  | -0.6913 | 0.3662  |                                    |

Table 1. results of MVA analysis at the four magnetosheath encounters.

Furthermore, I am concerned about the identification of KH vortices in the absence of low density plasma that is faster than the magnetosheath plasma (e.g., line 141+). This feature has been used to identify vortices using single spacecraft measurements. It may also be observed without vortices being present: the passage of a surface wave should suffice. But if vortices are present, then the feature should be observable too, and I cannot find any statement in Masson and Nykyri (2016) that would suggest the opposite. So the absence of that feature indicates, in my opinion, that the oscillations are rather related to a magnetopause surface wave rather than to KH vortices. Note that rotations in the bulk velocity, magnetic field deviations, and distortions of the magnetopause can also result from magnetopause surface waves (e.g., line 205).

We agree that the high-speed low-density feature is one of the typical characteristics of the K-H vortices and very useful in diagnosing the K-H vortices in single spacecraft measurements. It was a surprise to us that the high-speed low-density feature did not appear in the Ni-Vm plot in this

event. We used to estimate the magnetosheath velocity by drawing a horizontal line that is close to most of the magnetosheath intervals shown in panel 4 of figure 2. The horizontal line was at the velocity of about 180 km/s. Now, we re-estimate the magnetosheath velocity by averaging the TH-A measurements during the four magnetosheath intervals, with the more accurate velocity of about 134 km/s in the magnetosheath. Based on the new estimation, the high-speed low-density feature can be seen in the Ni-Vm plot, with more data points distributed in blue box (see the revised figure 5 below). The high-speed low-density feature can support the K-H vortices in this event.

Furthermore, the linear surface waves could not explain the fine structure of the observed first perturbation in TH-E observations, shown in the additional figure below, which is worthy of an independent article and we are further working on, with new results of the micro-physical process to transport solar wind into magnetosphere within the K-H vortices. The double peaks in the ion measurements can be caused either by second traversal of the non-linear K-H vortex or by the secondary substructure of the vortex. More details of the plasma transport in the K-H vortex will be revealed in another article in preparation.

Figure 5. The observed velocity along the tailward direction versus the ion density. Green dots are from TH-A observations and black dots from TH-E observations. The blue lines mark the high-speed and low density region possibly caused by the acceleration of the rotation.

---

## Author Response (AR2)

Dear Editor,

Thank you very much for your time spent on evaluating our article "Plasma transport into the duskside magnetopause caused by Kelvin-Helmholtz vortices as a response to the northward turning of the interplanetary magnetic field observed by THEMIS". We thank one of the referees who showed his/her agreement to the revised version of the article. We also thank the other referee who presented his/her lengthy comments and suggestions on our article. The comments and suggestions are very valuable and helpful to improve our scientific ideas and presentations in our article. We learned much from the comments that can help us not only in the current work, but also in the future work. Under the help of the comments and suggestions, we have revised the manuscript as follows:

1. The abstract and the summary have been rewritten by toning down the conclusions.

2. Some discussions of alternative explanations have been added to the text, as shown in the marked-up revision file.

3. Further check throughout the manuscript has been made in order to avoid possible and unexpected typos and incorrect statements.

We hope the revised version could satisfy both referees and the Editor.

**Reply to comments of Anonymous Referee #1**

We would like to thank the referee for the lengthy comments and suggestions on our manuscript. The comments and suggestions are very valuable and helpful to improve our scientific ideas and presentations in our article. We learned much from the comments that can help us not only in the current work, but also in the future work. We are responding the comments point-to-point in Blue words embraced in square brackets below your original comments.

First of all, I would like to thank the authors for their work to respond to all the reviewers' comments. They have taken all the comments into account. Nevertheless, I have to admit that I still have doubts about the identification of the KH vortices and about the interpretation of the measurements with respect to the plasma transport across the magnetopause, facilitated by the KHI. In my opinion, the paper should be published if all the conclusions drawn from the data and analysis are formulated in a much more careful and cautious way.

**Specific comments:**

The authors now state the eigenvalue ratios associated with the MVA; this is very good. It is, however, not surprising that those ratios are quite small, due to the small variations in the magnetic field across the magnetopause. Contrary to what is stated in the manuscript, the MVA results are considered reliable when the ratio is above 10, and acceptable when it is above 4. The latter condition only holds for 2 of the 8 analyzed intervals. Hence, local boundary normal directions obtained from MVA may well be considered only partially reliable or

even unreliable in this case. Unfortunately, these MVA-based directions are still the main argument for the KH vortex identification.

[Thank you for your endorsement on our adding the eigenvalue ratios of the MVA. As stated in the Table 1, the eigenvalue ratios are larger than 4 at the first pair of traversals of the magnetopause, and larger than 3 at two other single traversals. In previous research, the valve of the eigenvalue ratio was taken as 4 (e.g. Sergeev et. al. 2006). The Referee also said that it can be accepted if the ratio is larger than 4. This means that, at least, the calculated LMN coordinates are reliable at the first pair of traversals, indicating the formation of a vortex there. On the other hand, the low ratios mean only the failure of the MVA, but could not exclude the K-H vortices there. Only if the calculated LMN coordinates at one Pair of traversals of the magnetopause are reliable, the K-H vortex should have been generated. However, it should be admitted that the low ratios may degrade the convincibility of the results. We choose to describe the facts clearly in the text. ]

In the revised manuscript, some higher-speed lower-density plasma measurements are indeed identified, after lowering the magnetosheath velocity to 134 km/s by taking the average over some magnetosheath intervals. However, I would say that the original estimate of 180 km/s as magnetosheath velocity at the position of TH-A was more accurate, as it was probably observed further out in the "undisturbed" magnetosheath (see panels 4 and 5 of Figure 2). Note that the existence of high-speed low-density plasma does not ensure that the observations pertain to rolled-up vortices. The same plasma features can also come from magnetopause surface waves that are not amplified by the KHI. It is also not unimaginable that those non-KH surface waves may have fine structure yielding the double peaks observed by TH-E.

[Thank you for your agreement on the identified higher-speed lower-density plasma measurements. In this event, the northward IMF makes it difficult to identify the magnetopause. At the same time, it was lucky TH-A observed the magnetosheath during 4 intervals, which makes it possible to carry out not only the MVA calculations, but also to an estimate the magnetosheath velocity. Obviously, the observed velocity appeared different during different magnetosheath intervals. Under such circumstances, it is much more reasonable to calculate the averaged velocity than to estimate. And fortunately, the higher-speed lower-density feature was available if we take the average magnetosheath velocity.]

Consequently, the existence of KH vortices is suggested by not very reliable MVA-based directions and a few inconclusive high-speed low-density plasma observations. Rotation features in the bulk velocity and magnetic field deviations can also come from surface waves that are not subject to the KHI. The fact that the periodic magnetopause oscillations started with the northward turning of the IMF supports at least the assumption that the magnetopause surface waves were amplified/driven by the KHI – herein I agree with the authors. An argument against the existence of rolled-up vortices is the

observation location, far upstream of the terminator. Rolled-up vortices are known to form at and beyond the terminator; they should collapse further down the flank/nightside magnetopause. The manuscript claims that in this particular case the vortices form, fully developed, and collapse before even reaching the terminator.

[Thank you for your agreement with us that the periodic magnetopause oscillations started with the northward turning of the IMF supports at least the assumption that the magnetopause surface waves were amplified by the KHI. The surface waves might have pre-existed before the northward turning of IMF. The rolled-up vortices should occur seldomly before the terminator. But the past experience couldn't be used to exclude the possible K-H vortices as the deformation of the magnetopause at least into one vortex. The high-speed low-density plasma measurements were identified, although they are not so strong. Furthermore, some previous researches also mentioned a surprising observation of rolled-up vortices even at the dawnside magnetopause far upstream of the terminator (e.g. Lin et. al. 2014; and e.g. Grygorov et. al. 2016). The new observations will enrich our understanding of the K-H vortices.] Regardless of whether there are KH vortices or not, it is an interesting question if secondary processes at the magnetopause led to the transport of magnetosheath plasma into the magnetosphere. Contrary to what is stated in the response to the reviewers' comments letter and in the manuscript (e.g., line 178, line 200), the coexistence of hot and cold ions/electrons does not prove any local transport of particles into the magnetosphere. Coexisting plasma populations could have been already present as part of a pre-existing LLBL, which the authors admittedly cannot exclude. Consequently, in my opinion, there is neither "unambiguous" (original manuscript) nor "clear" (revised manuscript) evidence of plasma transport based on these "coexisting plasma" measurements; they do not provide evidence for plasma transport in this case. [We strongly agree with you that it is an interesting question if one or more certain secondary processes occurred at the magnetopause and led to the transport of magnetosheath plasma into the magnetosphere, regardless of whether there are K-H vortices or not. It is a great question, and it depends on how the secondary process works there. We hope our further investigation will answer it in another article in preparation. The coexisting of cold and hot plasmas have been identified in this event, hence we could exclude that it could be the intrinsic feature of the "pre-existing" LLBL. Nevertheless, it is still a question to identify the local transport at the LLBL. But compared with previous research results, such coexisting events identified by both electron and ion fluxes are strong evidence for the plasma transport in the LLBL. In the text, we have toned down the description per your suggestion. ]

Indirect evidence for plasma transport may only come from the evolution between periodic and dispersed magnetopause observations from TH-A to TH-E. The authors argue that the differences between the spacecraft come from the collapse of the KH vortices between them; this is possible but may not be most likely (see above). The same observations may potentially be more easily explained by different positions/distances of the respective spacecraft with respect to the magnetopause (wave), and by a pre-existing LLBL. Changing spacecraft distances could also explain the later periodic magnetopause oscillations observed by TH-E but not by TH-A. In this alternative scenario, in the absence of secondary processes at the magnetopause, local plasma transport would not be expected.

[It is a good idea that the evolution between periodic and dispersed magnetopause observations from TH-A to TH-E supply the indirect evidence for plasma transport. Thank you. In the text, we discussed more about possible alternative explanations such as the spacecraft's different distances to the magnetopause, or the intrinsic feature of the pre-existing LLBL as you suggested. In our further investigation, we hope to have the opportunity to show more details about the K-H vortex and its substructure, and furthermore, the possible mechanism as to how the vortex collapsed and led to local plasma transport into LLBL. ]

What can we conclude at the end? Are there KH vortices that form at the dayside magnetopause and collapse between TH-A and TH-E, dayside of the terminator? Maybe, maybe not. Most probably it can neither be fully proven nor excluded. Is there transport of magnetosheath plasma into the magnetosphere as a result of secondary processes at the magnetopause? Again, maybe, maybe not. It can neither be proven nor excluded. And here lies the essence of my criticism: The conclusions in the manuscript are formulated way too strongly, as if there were no possibility of doubt or alternative explanation. Should the manuscript the published, I would strongly encourage the authors to tone down all the conclusions and discuss possible alternative explanations.

[In this event, the periodical magnetopause oscillations were observed by TH-A and TH-E at the duskside magnetopause before the terminator. Although the rotational features in the bulk velocity, and the magnetic deviation could also be explained by surface waves, but the deformation of the magnetopause at least in one pair of magnetopause traversals and the high-speed low-density plasma measurements still indicate the generation of the K-H vortices. Since the evidence is not so convincing, we have toned down the conclusions according to your suggestion. By doing so, the event was described more moderately and more objectively. Thank you again.]

[revised manuscript text omitted]